# Time-Series Foundation Models for Forecasting Soil Moisture Levels in Smart Agriculture

Boje Deforce
boje.deforce@kuleuven.be
Research Center for
Information System Engineering
KU Leuven, Belgium

Bart Baesens
bart.baesens@kuleuven.be
Research Center for
Information System Engineering
KU Leuven, Belgium
Department of
Decision Analytics and Risk
University of Southampton, UK

Estefanía Serral Asensio
estefania.serralasensio@kuleuven.be
Research Center for
Information System Engineering
KU Leuven, Belgium

## ABSTRACT

The recent surge in foundation models for natural language processing and computer vision has fueled innovation across various domains. Inspired by this progress, we explore the potential of foundation models for time-series forecasting in smart agriculture, a field often plagued by limited data availability. Specifically, this work presents a novel application of `TimeGPT`, a state-of-the-art (SOTA) time-series foundation model, to predict soil water potential ($\psi_{\text{soil}}$), a key indicator of field water status that is typically used for irrigation advice. Traditionally, this task relies on a wide array of input variables. We explore `TimeGPT`'s ability to forecast $\psi_{\text{soil}}$ in: (*i*) a zero-shot setting, (*ii*) a fine-tuned setting relying solely on historic $\psi_{\text{soil}}$ measurements, and (*iii*) a fine-tuned setting where we also add exogenous variables to the model. We compare `TimeGPT`'s performance to established SOTA baseline models for forecasting $\psi_{\text{soil}}$. Our results demonstrate that `TimeGPT` achieves competitive forecasting accuracy using only historical $\psi_{\text{soil}}$ data, highlighting its remarkable potential for agricultural applications. This research paves the way for foundation time-series models for sustainable development in agriculture by enabling forecasting tasks that were traditionally reliant on extensive data collection and domain expertise.

## CCS CONCEPTS

• **Computing methodologies → Artificial intelligence**; **Machine learning**; • **Applied computing → Agriculture**.

## KEYWORDS

Foundation models, Time-series, Forecasting, Soil Water Potential, Smart Agriculture

**ACM Reference Format:**
Boje Deforce, Bart Baesens, and Estefanía Serral Asensio. 2024. Time-Series Foundation Models for Forecasting Soil Moisture Levels in Smart Agriculture. In *KDD '24 Fragile Earth Workshop*. ACM, New York, NY, USA, 7 pages.

## 1 INTRODUCTION

Recent years have witnessed a paradigm shift in artificial intelligence (AI) research, driven by the emergence of foundation models in natural language processing (NLP) [5] and computer vision (CV) [7]. These models, trained on massive datasets and capable of complex tasks, have spurred a wave of innovation across various domains. More recently, there has also been a rise of foundation models for time-series forecasting [2, 9]. For a full overview, we refer to [20]. Based on these recent developments, we explore the potential of foundation models for time-series forecasting in agriculture, specifically focusing on predicting soil water potential ($\psi_{\text{soil}}$), a key indicator of field water status.

Gaining insight into future $\psi_{\text{soil}}$-levels is important for optimizing irrigation scheduling, ensuring crop health and efficient crop management [10]. In turn, these optimized practices can directly contribute to the United Nations Sustainable Development Goals (SDGs), such as SDG-6 and SDG-12[1]. Recently, researchers have explored forecasting methods ranging from classic time-series and machine learning techniques [23] to more advanced deep learning-based methods like bi-directional Long Short-Term Memory networks (LSTMs) [8] and transformer-based approaches [6]. A common factor among these approaches is their substantial data requirements and the need for domain-specific knowledge to develop a successful combination of input variables (see Figure 1 – top). Yet, obtaining sufficient data in agriculture is not always straightforward [26].

To this end, this paper explores the applicability of foundation models for time-series forecasting in sustainable irrigation. We leverage the power of a recent SOTA foundation model, called `TimeGPT`, pre-trained on over 100 billion rows of financial, weather, Internet of Things (IoT), energy, and web data [9]. This approach bypasses the need for extensive training data by benefiting from the model's ability to capture intricate temporal relationships within the data due to its extensive pre-training on data from various domains (see Figure 1 – bottom). While foundation models are versatile by nature, they sometimes require fine-tuning when deployed for specific use-cases, which we also investigate in this work.

As such, our work contributes to the growing body of research exploring the application of foundation models beyond NLP and CV. By demonstrating the effectiveness of foundation models for $\psi_{\text{soil}}$ forecasting, we pave the way for their broader adoption in

---

[1]https://sdgs.un.org/goals

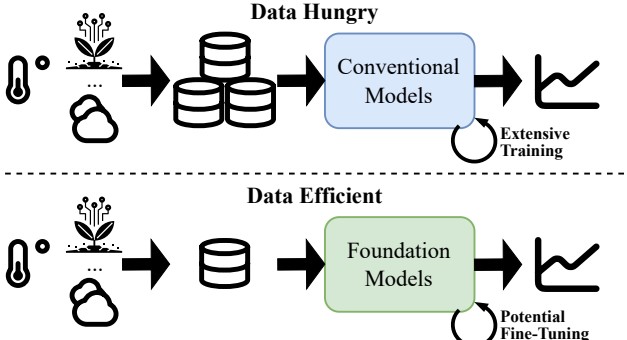

**Figure 1: Top: conventional approaches typically require large amounts of data to train a well-performing forecasting model, which is not always feasible in agriculture due to high costs involved in sensor setup, maintenance, ... Bottom: foundation models require no data at all for inference (in a zero-shot setting) or require far less data for fine-tuning compared to conventional approaches.**

agricultural decision-making, ultimately promoting sustainable water management practices. We summarize our main contributions as follows:

(1) Our work is the first to explore the use of time-series foundation models in an agricultural context, specifically for smart irrigation.

(2) We conduct an extensive comparison on real-world agricultural data against SOTA baselines, and include evaluations of both zero-shot and fine-tuned foundation model performance.

(3) We provide insights and future directions for developing foundation models tailored to agricultural time-series forecasting.

The rest of the paper is organized as follows. Section 2 presents the related work. Section 3 formalizes the forecasting problem and presents the settings of the study. Section 4 presents the results of the analysis comparing `TimeGPT` with SOTA approaches and a discussion. Finally, Section 5 provides a reflection on societal impact, followed by a conclusion and directions for further work in Section 6.

## 2 RELATED WORK

This section presents the most representative work related to the forecasting of $\psi_{soil}$ as well as the use of foundation models for time-series analysis.

### 2.1 Forecasting Soil Water Potential

Forecasting $\psi_{soil}$ is an important area of study, as part of the broader field of smart irrigation and smart agriculture. Proper insight into future $\psi_{soil}$-levels allows practitioners to schedule irrigation policies and monitor general crop health. Driven by a paradigm shift in industry [3] and academia [15] alike, recent literature has shown a growing interest in using deep learning techniques for forecasting.

The field of smart agriculture is no exception [16]. For example, the authors of [1, 24], used multi-layer perceptrons (MLP) and LSTMs to forecast soil moisture levels, with promising results given sufficient data. Several others have explored LSTM-based approaches due to their ability to capture temporal patterns. The authors of [8] employed bi-directional LSTMs to forecast up to 14 days ahead. Note that the use of a bi-directional LSTM may be impractical when not all variables are known 14 days in advance, as is often the case in real-world settings. In a more recent effort in [19], the authors combined LSTMs with an attention mechanism, improving performance and interpretability for forecasts ranging from one to seven days. Others utilized graph neural networks [30] or convolutional neural networks (CNNs) with gated recurring units [31] to capture the spatiotemporal relationships in soil moisture forecasting. In recent work, the authors of [6] investigated a transformer-based architecture to forecast $\psi_{soil}$ five days ahead, outperforming an LSTM baseline. They also showed that a global approach outperforms local approaches, a general trend in forecasting [3, 18].

However, a common denominator among these approaches is their reliance on substantial amounts of data for training, often spanning multiple years or seasons. This reliance presents a key challenge in situations where there is only limited data available, insufficient to train a model, while the practitioner would still benefit from obtaining predictions to make informed irrigation decisions. This limitation is a significant motivator for this work, as we advocate for the use of pre-trained foundation models that can provide useful predictions without training (in a zero-shot setting) or by fine-tuning them on a few samples of data. Notably, to the best of our knowledge, no previous works have considered foundation models for forecasting $\psi_{soil}$-levels.

### 2.2 Time-series Foundation Models

While the field of time-series foundation models is still relatively new, with one of the earliest examples in [22], a significant effort has been made to leverage insights from the explosion of literature on large language models (LLMs). For example, the authors of [22] showed that a pre-trained model could be successfully transferred across univariate time-series forecasting tasks without retraining (i.e., a zero-shot setting). One example [2] literally uses LLMs by using a quantization scheme to convert time-series into discrete tokens, leading to a zero-shot LLM-based forecasting model. For a full overview on recent efforts concerning foundation models for time-series, we refer the reader to [20]. Notably, two recent works stand out: `TimeGPT` [9] and `Lag-Llama` [25], as they allow both for zero-shot forecasting *and* fine-tuning [20]. As the former allows for the inclusion of exogenous variables, a potentially important factor in agriculture, we only consider `TimeGPT` in this work. Note that `TimeGPT` also showed superior results in [9] compared to its competitors.

## 3 METHODS

In what follows, we formalize the problem of $\psi_{soil}$ forecasting and describe the setting. Next, we give a detailed account of the models considered and provide details on the training and evaluation set-up.

## 3.1 Problem Description

Given a (potentially multivariate) time-series of input variables, we aim to forecast $\psi_{\text{soil}}$ up to 5 days ahead. Let $\mathbf{X}_t = (x_{t,1}, x_{t,2}, \ldots, x_{t,m})$ denote the vector of input variables at time $t$, where $m \geq 1$ is the number of variables. These input variables can include a range of environmental and meteorological factors such as temperature, rainfall, ... and previous $\psi_{\text{soil}}$ measurements (see Section 3.2). Our target variables are the $\psi_{\text{soil}}$ values for the next 5 days, denoted as $\mathbf{y}_{t+1:t+5} = (y_{t+1}, y_{t+2}, \ldots, y_{t+5})$. That is, a multi-horizon forecasting problem.

The objective is to develop a predictive model that uses the past values of $\mathbf{X}_t$ to forecast the $\psi_{\text{soil}}$ up to $h$ time-steps into the future, i.e. $\mathbf{y}_{t+1:t+h}$, where $h = 5$ days. The time-series data can be represented as a set of sequences $(\mathbf{X}_t^{(n)}, \mathbf{y}_{t+1:t+5}^{(n)})_{t=1}^{T_n}$ for $n = 1, 2, \ldots, N$, where $T_n$ is the number of time steps in the $n$-th time-series.

Formally, we seek to find a function $f$ such that:

$$\hat{\mathbf{y}}_{t+1:t+h}^{(n)} = f(\mathbf{X}_t^{(n)}, \mathbf{X}_{t-1}^{(n)}, \ldots, \mathbf{X}_{t-k+1}^{(n)}; \boldsymbol{\theta}^*) \tag{1}$$

where $\hat{\mathbf{y}}_{t+1:t+h}^{(n)} = (\hat{y}_{t+1}^{(n)}, \hat{y}_{t+2}^{(n)}, \ldots, \hat{y}_{t+5}^{(n)})$ are the predicted $\psi_{\text{soil}}$ values for the next $h$ for the $n$-th time-series, and $k$ is the number of past time steps considered by the model which is model-dependent. Note that Eq. (1) represents a (multi-horizon) *point* forecast. In the case of TimeGPT and some other baselines, a conditional *distribution* $\mathbb{P}(\mathbf{y}_{t+1:t+h}^{(n)}|\mathbf{X}_{t-k+1:t}^{(n)})$ is predicted. To find the optimal model parameters $\boldsymbol{\theta}^*$, we minimize the chosen loss function (which is model-dependent), $\mathcal{L}_{\text{trn}}$, during training as follows:

$$\boldsymbol{\theta}^* = \arg\min_{\theta} \mathcal{L}_{\text{trn}}(\hat{\mathbf{y}}, \mathbf{y}) \tag{2}$$

where $\hat{\mathbf{y}}$ are the predicted values and $\mathbf{y}$ are the true values across all $N$ time-series.

In the context of foundation models, the parameters $\boldsymbol{\theta}^*$ are obtained by pre-training on a massive dataset. Once trained, these parameters are frozen during inference when considering a zero-shot setting, meaning that they are not updated further based on new training data. This allows the model to make predictions on new inputs without altering its pre-learned parameters. Alternatively, such foundation models can be fine-tuned by unfreezing and updating (part of) the parameters during several training iterations.

For other models such as the baselines described in Section 3.4, a training subset of the $N$ time-series is used to find the parameters $\boldsymbol{\theta}$ that minimize the given loss function (cf. Eq. (2)). The amount of data required for training depends on the size of $\boldsymbol{\theta}$, with deep learning models (as described in Section 2) typically needing large datasets to achieve good performance. This requirement can be challenging to meet in an agricultural context, which is a key driver for considering foundation models that leverage pre-trained parameters for such applications.

## 3.2 Dataset

In this study, we use experimental data (obtained from [14]) from three commercial pear orchards during 2007 and 2009 in Belgium. We provide a brief description of each orchard below and provide a full overview of all variables in Table 1.

*Orchard A.* This orchard is located in Bierbeek, featuring 'Conference' pear trees grafted onto quince C rootstocks. These trees were spaced at 3.3 x 1 meters and were meticulously trained in an intensive V-system to optimize sunlight exposure and air circulation. The orchard's soil profile transitions from loam in the upper layer to sandy loam below, with an organic carbon content of 1% in the top 30 cm. The orchard underwent a regime of annual root pruning until 2006, followed by selective pruning in the years of the study.

*Orchard B.* Situated in Sint-Truiden, this orchard grew 'Conference' pear trees on quince Adams rootstocks, at intervals of 3.5 x 1.5 meters. Unlike Orchard A, these trees have never undergone root pruning and are trained in a free spindle system, which allows for a more natural growth pattern. The orchard has a loamy soil with slightly higher organic carbon content than Orchard A.

*Orchard C.* Located in Meensel-Kiezegem, Orchard C also featured 'Conference' pear trees on Quince Adams rootstocks, with the same planting pattern as Orchard B. This orchard is distinctive for its shallow groundwater table and slight slope, factors that significantly influence soil moisture dynamics and tree water uptake. Similar to Orchard A, selective root pruning was employed here in the study years.

In all orchards, standard commercial practices for pruning, disease control, fertilization, and mulching were consistently applied, ensuring that the variations observed in soil water status could be attributed to the differences in soil, rootstock, and specific management practices like root pruning. This experimental setup provides a comprehensive basis for comparing model performance.

## 3.3 Time-Series Foundation Models

Time-series foundation models – and foundation models more generally – are models that have been pre-trained on broad data at scale and are typically adaptable to a wide range of downstream tasks [4]. The success of foundation models is largely rooted in scaling laws suggesting that model performance (in terms of loss) scales as a power-law with model size, dataset size, and the amount of compute used for training [17]. An overview of foundation models for time-series is provided in Section 2.2. Here, we will consider the recently developed TimeGPT [9] due to its ability to generate zero-shot forecasts, while also allowing for fine-tuning and the inclusion of exogenous variables. Moreover, it achieved superior results compared to its competitors in [9]. TimeGPT is a generative pre-trained transformer model trained for time-series forecasting and anomaly detection. The model utilizes a transformer architecture [29], characterized by its encoder-decoder structure, where the encoder processes the input sequence of historical values $(\mathbf{X}_t^{(n)}, \mathbf{X}_{t-1}^{(n)}, \ldots, \mathbf{X}_{t-k+1}^{(n)})$ with local positional encodings, and the decoder followed by a linear layer generates the forecast $(\hat{y}_{t+1}^{(n)}, \hat{y}_{t+2}^{(n)}, \ldots, \hat{y}_{t+h}^{(n)})$ over the specified horizon $h$. Each layer within the architecture employs self-attention mechanisms to capture intricate temporal dependencies, enhanced by residual connections and layer normalization for stability and efficient gradient propagation [9, 29]. TimeGPT, as opposed to classic Transformers [29], uses CNN blocks. Without clear justification in [9], we assume this is due to their ability to capture spatiotemporal relationships

**Table 1: Overview of all variables and their description.**

| Variable | Description |
|---|---|
| Soil water potential ($\psi_{soil}$) – Target | The $\psi_{soil}$-values averaged per day. |
| Orchard name | Orchard name differentiates between orchards and (implicitly) their characteristics. |
| Soil texture | The texture of the soil at a 0-30cm depth. |
| Pruning treatment | Whether roots were pruned or not. |
| Irrigation treatment | Whether deficit irrigation was applied or not. |
| Measurement month | Measurement month. |
| Precipitation | Daily total precipitation. |
| Reference evapotranspiration | The reference evapotranspiration (ETo). |
| Irrigation amount | Amount of irrigation applied to a specific plot. |
| Soil temperature | Daily mean soil temperature around soil moisture sensors (measured by soil moisture sensor). |

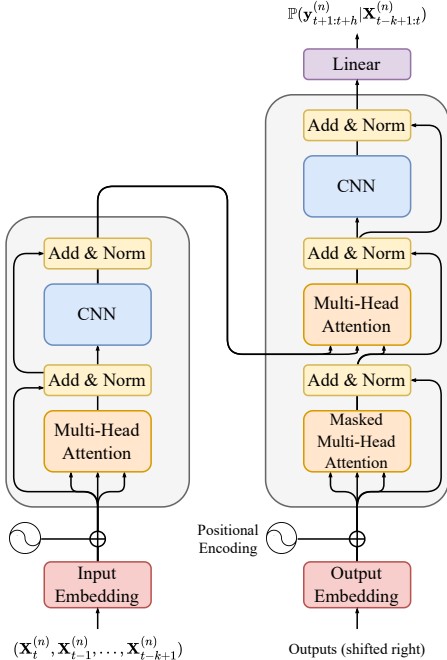

**Figure 2: Overview of the `TimeGPT` architecture based on [9]. Note that X can be univariate (containing only the history of the target), or multivariate (containing exogenous variables along with the history of the target).**

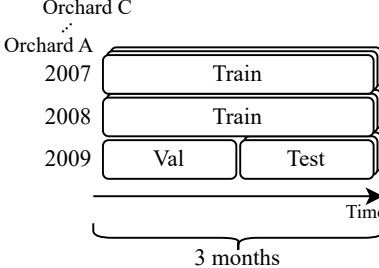

**Figure 3: Overview of the datasplits into training, validation, and test set. Final performance is reported on the test set.**

fine-tuned version using part of the training data, and (*iii*) a fine-tuned version using part of the training data while also including exogenous variables described in Table 1.

### 3.4 Baselines

We employed a range of baselines to compare `TimeGPT` against. First, we implemented a simple naive baseline, which propagates the last known value of $\psi_{soil}$ forward as the forecast, serving as a straightforward yet informative benchmark. Additionally, we included a Vector Autoregressive (VAR) model [27], leveraging its capability to capture linear interdependencies among multiple time-series. For more complex and non-linear relationships, we incorporated a Long Short-Term Memory (LSTM) network [11], which has demonstrated notable success in $\psi_{soil}$ forecasting (cf., Section 2) due to its proficiency in handling long-term dependencies and sequential data. Furthermore, we adopted the Temporal Fusion Transformer (TFT) [21], a state-of-the-art model that has also shown superior performance in $\psi_{soil}$ forecasting. The TFT combines the strengths of attention mechanisms and recurrent networks to effectively manage temporal dynamics and variable interactions. These diverse baselines enable a comprehensive assessment of `TimeGPT`'s performance across different modeling paradigms.

### 3.5 Training & Evaluation

To train the baseline models, the dataset is divided into training, validation, and test sets, taking into account the temporal and grouped

efficiently. A full overview of the `TimeGPT` architecture is provided in Figure 2. Note that `TimeGPT` provides a probabilistic forecast, rather than a point forecast. It does this by leveraging a conformal prediction framework [28]. Particularly, it estimates the model's error by performing a rolling forecast on the latest available data, before outputting the actual forecast along with its estimated error bounds. Lastly, in this work we consider three version of `TimeGPT`: (*i*) the zero-shot version (i.e., we use `TimeGPT` off-the-shelf), (*ii*) a

**Table 2: Overview of the results. The TFT performs the strongest, followed by `TimeGPT` with fine-tuning. While a simple naive method performs strong in comparison, Section 4.2 shows important qualitative differences compared to the TFT and `TimeGPT`. Best in bold, runner-up underlined.**

| Model | Exogenous included? | 5-day horizon | | | |
| | | MAE | | RMSE | |
| | | Mdn | IQR | Mdn | IQR |
|---|---|---|---|---|---|
| Naive | | 3.5083 | 6.0333 | 4.0472 | 6.8515 |
| ARIMA | | 3.7660 | 5.8742 | 4.4793 | 6.9607 |
| VAR | ✓ | 4.6148 | 7.0261 | 5.1908 | 7.8176 |
| LSTM | ✓ | 3.4926 | 5.9490 | 4.2354 | 6.8280 |
| TFT | ✓ | **2.7532** | 4.8498 | **3.3379** | 5.4107 |
| TimeGPT Zero-Shot | | 3.7421 | 7.0573 | 4.2979 | 7.4525 |
| TimeGPT Fine-Tuned | | 3.2056 | 4.7456 | 3.7350 | 5.8773 |
| TimeGPT Fine-Tuned Exo. | ✓ | 6.5171 | 6.2067 | 7.5461 | 7.5425 |

nature of the data to avoid any overlap or information leakage among these sets. Figure 3 illustrates how the data is segmented. Importantly, observations from the years 2007 and 2008 are designated solely for the training set ($n \approx$ 45k – 92%). This approach ensures that the model is trained on data spanning two full growing seasons, enabling it to accurately learn and account for the seasonal fluctuations present in the dataset. The validation set is used for model selection ($n \approx$ 2k – 4%). Finally, the test set is used as a representative unseen real-world case to evaluate the predictions on ($n \approx$ 2k – 4%). We calculate the mean absolute error (MAE) and root mean squared error (RMSE) across each horizon, and report the median across all time-series in the test set, along with their interquartile ranges (IQR).

Note that `TimeGPT`, in the zero-shot setting, does not require any training or validation data, as there is no training to be done. For fine-tuning `TimeGPT`, we use a subset of the training data and use 250 training iterations (which takes approx. one minute on CPU). Fine-tuning involves continuation of model training, starting from the pre-trained parameters [9].

## 4 EXPERIMENTS

In this section, we aim to answer the following questions:

- **Q1. Feasibility Assessment:** Is `TimeGPT` capable of forecasting $\psi_{\text{soil}}$?
- **Q1. Quantitative Analysis:** How does `TimeGPT` perform compared to other methods?
- **Q1. Qualitative Analysis:** Can `TimeGPT` pick up on $\psi_{\text{soil}}$ patterns?

## 4.1 Quantitative Results

Table 2 shows the quantitative results for `TimeGPT` and all baselines. A few observations stand out. For example, the TFT performs the best overall. This is not entirely surprising given its strong performance across various domains [12, 13, 32]. We believe that its strong performance is also due to the explicit encoding of static information (e.g., orchard characteristics), which is not present in any of the other models. Interestingly, the second best-performing model is `TimeGPT`, fine-tuned using only the history of the target itself. This

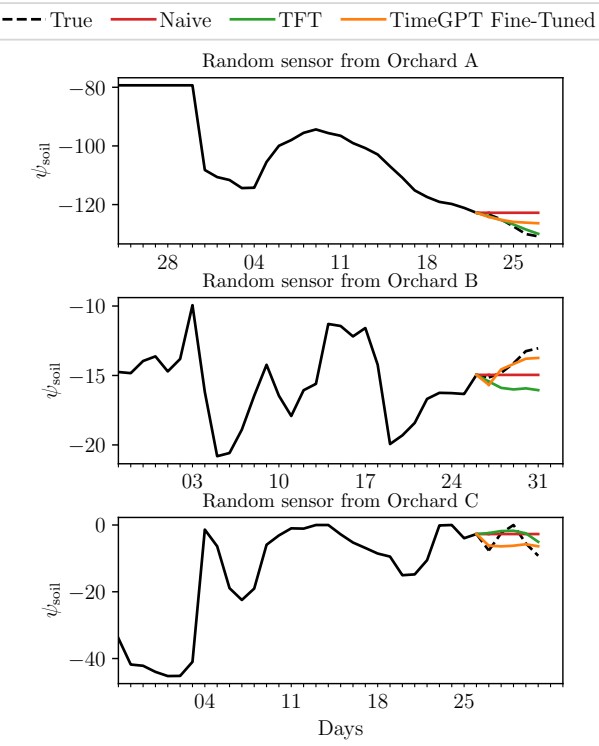

**Figure 4: Forecasts from the best performing models and the naive for three randomly sampled sensors across different orchards. The error bands for `TimeGPT` and the TFT are omitted for clarity.**

is an important observation as this required significantly less data and effort compared to training the TFT from scratch. As the authors of [9] indicate, implementation complexity and computation cost[2] are key factors for practical adoption. One can easily imagine that practitioners in agriculture may favor a frictionless solution as opposed to training a complex model like the TFT from scratch. Surprisingly, adding exogenous data while fine-tuning `TimeGPT` leads to much worse results. Due to the black-box nature of `TimeGPT`, it's unclear why this is happening. We conjecture that the complex agricultural relationships among agricultural variables are not well represented in the distribution of the data used during pre-training of `TimeGPT`. Of course, the seemingly strong performance of the naive model, cannot be ignored and is discussed in detail in the next section.

## 4.2 Qualitative Results

The relatively low error of the naive baseline is mainly due to strong correlations between the last known value and future values as shown in Figure 5. However, such naive forecasts are not useful to the practitioner as they do not contain any new information. This becomes further apparent in Figure 4. Here, we compare the

---

[2]Note that the computational cost is now shifted to pre-training

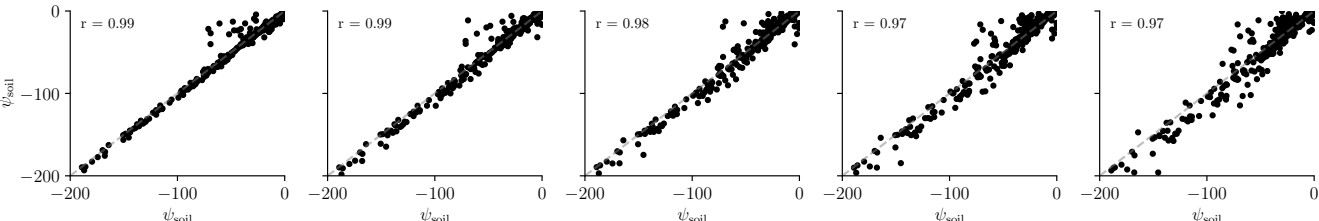

**Figure 5: Correlation plots across the forecasting horizon $h$ representing the correlation between $\psi_{\text{soil}}$ at time $t$ and $\psi_{\text{soil}}$ at $t + 1$ to $t + 5$. Note how the error becomes larger as time moves on.**

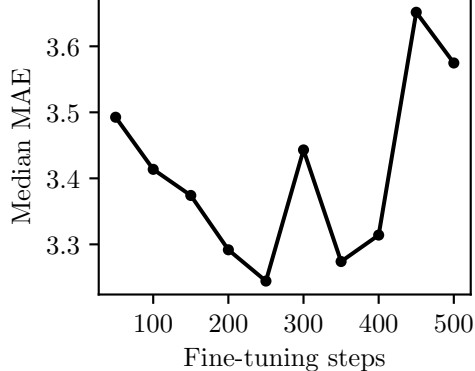

**Figure 6: Overview of how the MAE (median of MAE across multi-horizon forecasts across all time-series in test set) evolves as the number of fine-tuning steps of `TimeGPT` grows. Here, `TimeGPT` is considered without exogenous data.**

naive and the two best performing models (TFT and `TimeGPT` fine-tuned) on their forecasts for three randomly sampled sensors across the different orchards. Observe that the naive model, by definition, does not provide useful insights, particularly as time progresses. Consequently, the relatively strong performance of the naive model can be attributed to the fact that $\psi_{\text{soil}}$ remains relatively close to the last known value. From an agricultural perspective, this phenomenon is logical, as soil moisture, especially in wet climates, can take several days to dissipate depending on root distribution and other field characteristics. Meanwhile, the TFT shows strong performance in some scenarios (e.g., top), but lacks in others (e.g., middle). Similar observations are made for `TimeGPT`. Nonetheless, the results from Table 2 and Figure 4 are promising to consider `TimeGPT` as an enabler in smart agriculture. Furthermore, the level of fine-tuning of `TimeGPT` should also be considered. Figure 6 shows the importance of carefully choosing the number of fine-tuning steps. When the fine-tuning steps are low (the default is 10), the model is seemingly underfitting. Conversely, allowing for too many training steps leads to overfitting [9]. Regardless, this is essentially the only *hyperparameter* that needs tuned, as opposed to a panoply of hyperparameters in the TFT [21].

## 5 SOCIETAL IMPACT

The application of foundation models like `TimeGPT` in agriculture, despite not yet achieving optimal performance, holds significant promise for advancing sustainable agricultural practices in line with the SDGs. Specifically, leveraging such models can contribute to SDG-6 (Sustainable Development) by enabling more precise soil moisture forecasting, which supports efficient water use and can help improve crop yields.

Additionally, the reduced computational requirements and ease of fine-tuning `TimeGPT`-like models align with SDG-12 (Responsible Consumption and Production), promoting the adoption of resource-efficient technologies in agriculture. By providing actionable insights with minimal data and computational overhead, these models can facilitate more resilient and sustainable agricultural systems.

Moreover, the scalability and adaptability of foundation models can aid in addressing climate variability and environmental challenges, supporting SDG-13 (Climate Action). While the performance of `TimeGPT` in agricultural forecasting is still evolving, its potential to enhance sustainable farming practices and contribute to global food security and environmental stewardship should be considered. This opens doors to innovative, data-driven approaches that empower farmers with the tools needed for sustainable and resilient agriculture.

## 6 CONCLUSION

In this work, we evaluated the feasibility and performance of `TimeGPT` for forecasting soil moisture ($\psi_{\text{soil}}$) in an agricultural context, benchmarking it against several baseline models, including the state-of-the-art TFT. Our results demonstrate that `TimeGPT` is indeed capable of forecasting $\psi_{\text{soil}}$, with notable performance when fine-tuned solely on the target variable's history. This approach required significantly less data and effort compared to training the TFT from scratch, making `TimeGPT` a viable option for practical applications due to its lower implementation complexity and computational cost. Though, it should be noted that performance of `TimeGPT` is still sub-par compared to the TFT.

Surprisingly, incorporating exogenous data during `TimeGPT`'s fine-tuning resulted in poorer performance, potentially due to the pre-training data distribution not adequately capturing the complex relationships between agricultural variables. This highlights a need for future research to (*i*) explore `TimeGPT`'s performance on other agricultural datasets or (*ii*) better align pre-training data with

specific agricultural contexts. Despite the naive model's relatively strong performance, which can be attributed to the persistence of soil moisture values, its lack of actionable insights limits its utility for agricultural decision-making.

Qualitative analysis revealed that both the TFT and `TimeGPT` have strengths and weaknesses across different scenarios. The critical role of fine-tuning for `TimeGPT` was evident, with both notions of underfitting and overfitting observed depending on the number of fine-tuning steps, emphasizing the need for tuning of this hyperparameter. However, `TimeGPT` requires tuning of far fewer hyperparameters compared to complex models such as the TFT.

In conclusion, `TimeGPT` shows promise as a valuable tool for smart agriculture, with its ease of fine-tuning and lower computational demands, thereby directly and indirectly contributing to several SDGs. Future work should focus on optimizing the integration of exogenous data and refining the pre-training or fine-tuning process to further enhance `TimeGPT`'s performance in complex agricultural environments. Other competitive foundation models could also be considered.

## ACKNOWLEDGMENTS

Funding: This work was supported by the Research Foundation - Flanders (FWO) [grant number G0C6721N].

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
