# OpenReview forum: "Time-Series Foundation Models for Forecasting Soil Moisture Levels in Smart Agriculture"
_KDD.org/2024/Workshop/Fragile_Earth — Fragile Earth FullPresentation_

### Official Review · Reviewer_LjNg · 2024-07-08
**Time-Series Foundation Models for Forecasting Soil Moisture Levels in Smart Agriculture**

**Rating:** 7
**Confidence:** 5

**Review:**

This paper presents an interesting analysis of using off the shelf TimeGPT time-series foundational model to predict soil water potential that is typically used for irrigation advice. The authors present TimeGPT's prediction on soil water potential in (a) zero shot setting (b) fine-tuned setting relying solely on historic measurements and (iii) fine-tuned setting with additional features.

I have the following comments.

1. The details of fine tuning is missing. Did it involve just retraining using the embeddings from TimeGPT? In section 3.5, it was mentioned to look at section 4.2. However, I couldn't find the details.
2. It is interesting to see from Table2, that TimeGPT Fine-Tuned with out any additional features performed better than using exogenous features.
3. Also it is interesting to see that off-the-shelf TFT performed over TimeGPT foundational model.
4. For Figure 5, which model's results are presented -- TFT, Fine-tuned or Fine-Tune w/ exogenous features?

---

### Official Review · Reviewer_qU1s · 2024-07-13
**Review of Time-Series Foundation Models for Forecasting Soil Moisture Levels in Smart Agriculture**

**Rating:** 8
**Confidence:** 4

**Review:**

Summary:

In this paper, the authors present a novel application of TimeGPT, a novel time-series foundation model, to predict soil water potential which can be used for irrigation advice. In experiments, the authors compare TimeGPT’s performance to established state-of-the-art baseline models for forecasting and the proposed TimeGPT achieves competitive forecasting accuracy.

Strengths:

+ This paper is well-written and the improvement of the work over the previous state-of-the-art is very impressive.
+ Simplicity of the proposed method can be considered as a strength as it allows easier reproducibility.

Weaknesses:

- The authors may consider exploring different horizons and providing additional evaluation metrics (e.g., MAPE and etc).

---

### Official Review · Reviewer_ZFb8 · 2024-07-15
**review for "Time-Series Foundation Models for Forecasting Soil Moisture Levels in Smart Agriculture"**

**Rating:** 7
**Confidence:** 4

**Review:**

The authors explore the potential of time series foundation models for forecasting in agriculture. They applied TimeGPT to predict soil water potential up to 5 days into the future with both zero-shot and fine-tuned models using real-world data from three commercial pear orchards and compared against SOTA baselines. Compared to these baselines, foundation models do not require a large amount of data, which is more practical in data-scarce settings.

Results show that TimeGPT achieves competitive forecasting accuracy after fine-tuning with only historical soil water potential data. This research is novel and meaningful, as it opens up research avenues in using foundation models to build more sustainable agricultural systems, which requires less effort compared to other ML approaches.

Results in the paper showed that adding exogenous data for fine-tuning led to much worse results in prediction. To further understand the reason, the authors could try removing exogenous variables that are constant and non-numerical (such as. orchard name), or to experiment with one exogenous variable at a time, to understand the impact of each variable on the prediction. Additionally, the authors used a train-validation-test split on the time series data for each orchard currently. Another interesting experiment could be to fine-tune the model on two orchards and test the prediction accuracy on the third orchard, to evaluate the generalization ability of TimeGPT among the same prediction task after fine-tuning. Moreover, in Figure 6, it is expected that the MAE increases as the number of fine-tuning steps grows. However, the MAE increases drastically at around 300 steps but decreases at around 350 steps, before it increases again. It would be great if the authors could provide possible reasons for this unexpected behavior.

---

### Decision · Program_Chairs · 2024-07-24

Accept (Full Presentation)